


# 1 Upper stratospheric ClO and HOCl trends (2005–2020): Aura
# 2 Microwave Limb Sounder and model results

Lucien Froidevaux[1], Douglas E. Kinnison[2], Michelle L. Santee[1], Luis F. Millán[1], Nathaniel J. Livesey[1],
William G. Read[1], Charles G. Bardeen[2], John J. Orlando[2], and Ryan A. Fuller[1]
[1] Jet Propulsion Laboratory, California Institute of Technology, Pasadena, California, USA
[2] National Center for Atmospheric Research, Boulder, Colorado, USA
*Correspondence to*: Lucien Froidevaux (lucienf@jpl.nasa.gov)
**Abstract.** We analyze Aura Microwave Limb Sounder (MLS) monthly zonal mean time series of ClO and HOCl
between 50ºS and 50ºN to estimate upper stratospheric trends in these chlorine species from 2005 through 2020.
We compare these observations to those from the Whole Atmosphere Community Climate Model version 6
(WACCM6), run under the specified dynamics configuration. The model sampling follows the MLS coverage in
space and local time. We use version 5 MLS ClO zonal mean daytime profiles and similarly binned daytime ClO
model profiles from 32 to 1.5 hPa. For MLS HOCl, we use the version 5 offline product derived from daily zonal
mean radiances rather than averaged Level 2 profiles; MLS HOCl is scientifically useful between 10 and 2 hPa,
and the HOCl monthly zonal means are separated into day and night for comparison to WACCM6. We find good
agreement (mostly within ~10%) between the climatological MLS ClO daytime distributions and the model ClO
climatology for 2005–2020. The model HOCl climatology, however, underestimates the MLS HOCl climatology
by about 30%. This could well be caused by a combination of fairly large systematic uncertainties in both the
model-assumed rate constant for the formation of HOCl and the MLS HOCl retrievals themselves.
The model daytime ClO trends versus latitude and pressure agree quite well with those from MLS. MLS-
derived near-global upper stratospheric daytime trends between 7 and 2 hPa are –0.73 ± 0.40 %yr$^{-1}$ for ClO and –
0.39 ± 0.35 %yr$^{-1}$ for HOCl, with 2σ uncertainty estimates used here. The corresponding model decreases are
somewhat faster than observed (although the difference is not statistically significant), with trend values of
–0.85 ± 0.45 %yr$^{-1}$ for ClO and –0.64 ± 0.37 %yr$^{-1}$ for HOCl. Both data and model results point to a faster trend
in ClO than in HOCl. The MLS ClO trends are consistent with past estimates of upper stratospheric ClO trends
from satellite and ground-based microwave data. As discussed in the past, trends in other species (in particular,
positive trends in $CH_4$ and $H_2O$) can lead to a ClO decrease that is faster than the decrease in total inorganic



chlorine. Regarding trends in HOCl, positive trends in $HO_2$ can lead to a faster rate of formation for HOCl as a
function of time, which partially offsets the decreasing trend in active chlorine.
The decreasing trends in upper stratospheric ClO and HOCl provide additional confirmation of the
effectiveness of the Montreal Protocol and its amendments, which have led to the early stages of an expected
long-term ozone recovery from the effects of ozone-depleting substances.
**1 Introduction**
Changes in the gaseous chlorine content of the atmosphere have been scrutinized since the late 1970s, when
prescient warnings (Molina and Rowland, 1974) were made regarding likely threats to the Earth's stratospheric
ozone ($O_3$) layer from the decomposition of various chlorofluorocarbons (CFCs) emitted at the surface by human
industrial activities. These threats carried human health implications as a result of increased ultraviolet (UV)
radiation at the surface, which would follow from reductions in UV absorption by stratospheric ozone. Various
measurements of the abundances of different chlorine species in the stratosphere followed these early years of
concern regarding expected declines in global ozone. Early balloon-borne observations of chlorine monoxide
(ClO) radicals in the upper stratosphere (Anderson et al., 1977; Waters et al., 1981) confirmed the predicted
importance of gas-phase reactions (involving ClO, Cl, $O_3$, and O) on upper stratospheric ozone abundances. Since
the 1987 Montreal Protocol and its subsequent amendments, established to strongly reduce worldwide surface
emissions of halogenated compounds harmful to the ozone layer, both the tropospheric and stratospheric chlorine
budgets have been carefully studied and monitored by the atmospheric science community. This was motivated
by enhanced concerns regarding ozone decreases in the lower stratosphere, after the discovery of the seasonal
appearance of an ozone hole over Antarctica (Farman et al., 1985).
Studies of interannual and longer-term changes in stratospheric chlorine species were carried out by ground-
based (column) measurements of HCl and $ClONO_2$ at infrared wavelengths (Rinsland et al., 2003; Kohlhepp et
al., 2011; Mahieu et al., 2014). Near-global stratospheric chlorine changes have also been tracked by satellite
measurements of HCl. Indeed, this chlorine reservoir species at high altitude (near 50 km) accounts for the vast
majority of $Cl_y$ (total inorganic chlorine), based on past measurements of the stratospheric chlorine budget by
Zander et al. (1992) and Nassar et al. (2006). Froidevaux et al. (2006) also discussed model results regarding the
contribution of upper stratospheric HCl to $Cl_y$ and described measurable decreases in HCl (and by inference, in
$Cl_y$) from mid-2004 to early 2006, based on changes in Aura MLS profiles. The rather fast rise in chlorine from
the 1980s to the late 1990s (with increases of more than 55%) was followed by a slower rate of decrease, as





expected from model calculations. Stratospheric chlorine follows the overall tropospheric trends with about a 5-
year delay, which accounts for transport and mixing of tropospheric compounds into the stratosphere (as discussed
by Anderson et al., 2000, Waugh et al., 2001, and others).
Changes in chlorine source gases at the surface, as well as changes in stratospheric chlorine species, have been
updated and documented regularly in quadrennial reports (see WMO, 2018). Based on such analyses, stratospheric
HCl has been decreasing over the past two decades by about 0.5–1%yr$^{-1}$. This includes results from ground-based
infrared measurements, as well as from near-global upper stratospheric HCl measurements by the Atmospheric
Chemistry Experiment Fourier Transform Spectrometer (ACE-FTS) (see Bernath and Fernando, 2018). These
results are consistent with surface total chlorine trends, based on in situ sampling of a large number of source
species by ground-based networks (Engel and Rigby et al., 2018), so that there is a good corroboration of the
effectiveness of the Montreal Protocol and its amendments, except for some recent departures from expectations
for the evolution of CFC-11 (Montzka et al., 2018). Ground-based microwave measurements of stratospheric ClO
profiles over the past two decades have also made valuable contributions to these long-term chlorine composition
records. This includes trend results for upper stratospheric ClO over Hawaii (Solomon et al., 2006; Connor et al.,
2013) as well as for the more variable lower stratosphere over Antarctica (Nedoluha et al., 2016). These findings
corroborate the longer-term decreasing trends in HCl (and Cl$_y$), although dynamical variability on timescales of
5–7 years complicates trend detection (e.g., for HCl) in the lower stratosphere (Mahieu et al., 2014; Strahan et al.,
2020); this variability and its causes are still under investigation in the community.
Here, we provide an analysis of upper stratospheric trends in near-global ClO and hypochlorous acid (HOCl).
These two chlorine species have been measured by the Aura Microwave Limb Sounder (MLS) globally on a near-
daily basis since its launch in 2004. An analysis of their trends falls within the general theme of confirming that
the Montreal Protocol has been able to significantly reduce the threat of stratospheric chlorine to global ozone.
The MLS measurements of upper stratospheric ClO and HOCl have taken on a larger role, in light of the fact that
MLS lost the capability of obtaining trend-quality data on upper stratospheric HCl after a hardware issue in early
2006 (see Livesey et al., 2020). The lower stratospheric HCl measurements have continued through the use of
radiances from an adjacent MLS measurement band (see also the lower stratospheric MLS HCl comparisons to
model results by Froidevaux et al., 2019). In Section 2, we describe the observations, model simulations, and
methods of analysis for this work. Section 3 focuses on the trend results for ClO and HOCl, while Section 4
provides a discussion in the context of broader trends in upper stratospheric species. Our conclusions are
summarized in Section 5.



## 2 Observations, model simulations, and analysis methods

In this work, we analyze temporal changes in upper stratospheric ClO and HOCl abundances, based on continuous MLS observations of both species from 2005 through 2020. We compare these observational results to those from a state-of-the-art chemistry climate model for the same time period.

### 2.1 Observations

The primary datasets used in this analysis come from 16 full years (2005 through 2020) of global measurements performed by Aura MLS. The MLS antenna scans the atmospheric limb as the Aura satellite orbits the Earth in a near-polar sun-synchronous orbit; the instrument measures thermal emission (day and night), using microwave radiometers operating at frequencies near 118, 190, 240, and 640 GHz, as well as a 2.5 THz module to measure OH (during the early part of the mission only). MLS has been providing a variety of daily vertical stratospheric temperature and composition profiles (~3500 profiles per day per product), with some measurements extending down to the upper tropospheric region, and some into the upper mesosphere or higher. We rely here mainly on the upper stratospheric MLS measurements of ClO and HOCl, obtained from 640-GHz radiometer data. Specifically, ClO and HOCl emissions are obtained from lines centered at 649.5 and 635.9 GHz, respectively; Waters at al. (2006) have provided an overview of the MLS instrument and its measurements, along with some sample spectra, and Read et al. (2006) have described the simulated forward model and related spectra. The MLS retrievals use an optimal estimation approach (Rodgers, 2000), with MLS-specific details provided by Livesey et al. (2006); there is no assumption of atmospheric homogeneity along the line of sight (see Livesey and Read, 2000), and the MLS retrievals make use of the instrument's views (which are all along the line of sight) during multiple consecutive MLS antenna scans of the Earth's limb. Data users interested in MLS data quality and characterization, estimated errors, and related information, should consult Livesey et al. (2020), the latest update to the MLS data quality document.

In this work, we use the latest data version from MLS, namely version 5.0 (or v5). The single-profile precision (1$\sigma$ random uncertainty) is ~0.1 ppbv for the ClO retrievals in the region between 32 and 1.5 hPa that we focus on here; the vertical resolution of the ClO measurements is about 3–4 km. For our analyses of daytime MLS ClO monthly zonal means in 5° latitude bins, the more relevant precision for averaged upper stratospheric values drops to about 0.5–5%. In addition, the methodology used by the MLS team to assess the aggregate effects of simulated errors in various input parameters on the measurement retrievals (see Livesey et al., 2020) leads to systematic uncertainties of order 0.02–0.1 ppbv for upper stratospheric ClO, which translates to about 5–100% for ClO,



depending on whether one considers the peaks of the distributions (for the smaller uncertainty values) or regions
away from these peaks. The standard MLS data quality screening methodology (see the above reference) has been
applied to all Level 2 ClO profiles, prior to averaging into monthly zonal means.

For the MLS HOCl data, we have used an offline retrieval product that shows similar results as the averaged

Level 2 profiles, but with somewhat smaller variability. This product is created offline (i.e., after the daily
processing of incoming MLS data) by averaging daily Level 1 spectra before performing the retrievals of mean
daily profiles, which are then averaged for this work into either day or night monthly zonal means. The offline
retrieval technique follows the overall MLS retrieval methodology described by Livesey et al. (2006), except it is
a one-dimensional type of retrieval (as it is not used for line-of-sight 'chunks' of profiles like the Level 2
'tomographic' approach). Moreover, the radiances that are used as part of the averages correspond to profiles for
which the temperature and ozone retrievals in Level 2 have passed the standard retrieval criteria for good quality
data. This methodology is the same as that used for the MLS offline retrievals of BrO and $HO_2$, which are also
considered to be MLS "noisy products", based on their single-profile precision values (see Millán et al., 2012,
2015, for BrO and $HO_2$, respectively). These averaged offline products can be more stable and scientifically useful
over a wider vertical range than averages of the MLS Level 2 standard products (although the wider vertical range
only holds for $HO_2$). Also, the latitude grid spacing for the MLS offline HOCl product (as for the other offline
products mentioned above) is 10°, rather than the 5° used for ClO and other standard MLS retrieval products. We
have used the precision and accuracy HOCl estimates from the standard Level 2 MLS product, as we expect
similar uncertainties (or possibly better) for the offline HOCl product. The MLS HOCl precision for (day or night)
10° monthly zonal means is typically less than 5-10 pptv (or roughly 5–20%). Systematic uncertainties are
estimated to be 40–80 pptv for HOCl, or about 25–100%. The more limited useful vertical range for MLS HOCl
is 10 to 2 hPa, and the HOCl profiles have a vertical resolution of only 5–6 km.

We also make use of upper stratospheric data from ACE-FTS, which was launched in 2003 as part of the

Canadian SCISAT mission. The instrument uses the solar occultation technique and gathers measurements in the
infrared region (at 750–4400 $cm^{-1}$, with a spectral resolution of 0.02 $cm^{-1}$). The ACE-FTS sampling is skewed
towards middle to high latitudes, with many fewer profiles per day (per species) than obtained from MLS (30
from ACE-FTS versus ~3500 from MLS). ACE-FTS has provided a wealth of constituent profile measurements
over basically the same period as Aura MLS (see the overview by Bernath et al., 2017); we use some ACE-FTS
trend results to obtain a broader description and understanding of chlorine species trends in the upper stratosphere.
We have used ACE-FTS data version 4.1 in the analyses presented here; see Boone et al. (2020) and references
therein for detailed information on the ACE-FTS retrievals. We have removed the largest outliers in the ACE-



FTS data by using the prescription regarding data flags from Sheese et al. (2019), although this data screening
makes essentially no difference to the near-global upper stratospheric data averages and related trend results in
this work.
**2.2 Model simulations**
The model used here is the Whole Atmosphere Community Climate Model version 6 (WACCM6), a
component of the Community Earth System Model 2 (CESM2), configured to use specified dynamics as described
by Gettelman et al. (2019). These authors showed that this chemistry climate model reproduces many modes of
variability, as well as trends, in the middle atmosphere. WACCM6 is the "high-top" version of the Community
Atmosphere Model, version 6 (CAM6; Danabasoglu et al., 2019). CAM6 includes updated representations of
boundary layer processes, shallow convection, liquid cloud macrophysics, and two-moment cloud microphysics
with prognostic cloud mass and concentration. This version of CAM6 uses a finite volume dynamical core (Lin,
2004). The horizontal resolution is 0.95˚ latitude x 1.25˚ longitude. The model has 88 levels with a vertical range
from the surface to the lower thermosphere. The vertical resolution in the lower stratosphere ranges from 1.2 km
near the tropopause to ~2 km near the stratopause.
The WACCM6 model represents chemical processes from the troposphere into the lower thermosphere. The
chemical scheme includes the $O_x$, $NO_x$, $HO_x$, $ClO_x$, and $BrO_x$ chemical families, along with $CH_4$ and its
degradation products. This scheme also includes primary non-methane hydrocarbons and related oxygenated
organic compounds. The chemical processes have evolved from previous versions and are summarized in detail
by Emmons et al. (2020).  Reaction rates follow the JPL 2015 recommendations (Burkholder et al., 2015). The
chemical scheme also includes a new detailed representation of secondary organic aerosols (SOAs), based on the
"simple Volatility Basis Set" approach (Tilmes et al., 2019). WACCM includes a total of 231 species and 583
chemical reactions broken down into 150 photolysis reactions, 403 gas-phase reactions, 13 tropospheric, and 17
stratospheric heterogeneous reactions. The photolytic reactions are based on both inline chemical modules and a
lookup table approach (Kinnison et al., 2007).
The model scenario used here is based on historical forcings (and recent updates) from the Climate Model
Intercomparison Project – Phase 6 (Meinshausen et al., 2017). These include greenhouse gases ($CH_4$, $N_2O$, and
$CO_2$) and organic halogens ($CH_3Cl$, $CH_3CCl_3$, $CCl_4$, CFC-11, CFC-12, CFC-113, CFC-114, CFC-115, HCFC-22,
HCFC-141b, HCFC-142b, $CH_3Br$, halon-1211, halon-1301, halon-2402, $CHBr_3$, and $CH_2Br_2$). CMIP6
specification of $NO_x$ emissions from medium energy electrons (MEEs), solar proton events (SPEs), and galactic
cosmic rays (GCRs) is also included. The 11-year solar cycle variability is taken from the Naval Research



Laboratory's (NRL) solar variability model, referred to as the NRL Solar Spectral Irradiance version 2 (NRLSSI2;
Coddington et al., 2016). The volcanic $SO_2$ emissions (used in the sulfate aerosol density calculation) are derived
for each volcanic eruption using the Neely and Schmidt (2016) database updated through the year 2020. This work
uses the specified dynamics (SD) option (Lamarque et al., 2012), where reanalysis temperature, zonal and
meridional winds, surface stress, surface pressure, and surface latent and sensible heat are used to nudge the model
state, thus affecting parameterizations controlling boundary layer exchanges, advective and convective transport,
and the hydrological cycle. This model's dynamical constraints, including the Quasi-Biennial Oscillation (QBO),
arise from meteorological fields provided by the Modern-Era Retrospective analysis for Research and
Applications Version 2 (MERRA-2; Gelaro et al., 2017), and the nudging approach is described by Kunz et al.
(2011). The model meteorological fields are nudged from the surface to 50 km; above 60 km, these fields are fully
interactive, with a linear transition in between. The model nudging time constant is 50 hours. Model results are
obtained from a simulation that, originally, started in 1980 and ended in 2014 (Gettelman et al., 2019); it was later
augmented with runs through 2020. After 2014, the greenhouse gas and organic halogen inputs follow the CMIP6
SSP2-45 scenario (O'Neill et al., 2016; Riahi et al., 2017), the SPEs are derived from the Geostationary
Observational Environmental Satellites (GOES) proton fluxes (Jackman et al., 2008), and the MEEs and GCRs
are based on the CMIP6 pre-industrial control.

In terms of sampling, the flexibility of WACCM allows for a choice of profiles for local time and spatial
coincidences as close as possible to each MLS profile, using the roughly $1° \times 1°$ model bin that includes a given
data location for a model local time that falls within 15 minutes of the MLS local time, and binned according to
day or night criteria. The model's daily zonal mean profiles (sampled following the MLS locations and local
times) are interpolated (as a function of $log(p)$, where $p$ is pressure) to the MLS retrieval grid points; for ClO and
HOCl, this grid is defined by a stratospheric subset of $p(n) = 1000 \times 10^{-n/6}$, in units of hPa, where $n$ is the pressure
level index.
**2.3 Analysis methods**

We have used solar zenith angles less than 90° or larger than 100° to separate daytime from nighttime values,
respectively, for both MLS and model profiles; after this selection, monthly zonal means were created.

In terms of trend analyses, we follow the approach for MLS data and model trends discussed by Froidevaux et
al. (2019), namely a multivariate linear regression (MLR) method, in order to fit the monthly zonal mean time
series from both MLS and the model. We refer the reader to Appendix (A3) of the above reference for more details


regarding the regression model, which includes commonly used functional terms, namely a linear trend and a
constant term, cosine and sine functions with annual and semi-annual periodicities, as well as functions describing
variations arising from the QBO and the El Niño / southern oscillation (ENSO); ENSO plays a large role (in
comparison to the QBO) only in the lower stratosphere (e.g., Randel and Thompson, 2011). Here, we also include
a fitted component that follows variations in solar radio flux (at 10.7 cm), F10.7, based on the Canadian solar
measurements described by Tapping (2013). For the trend uncertainty estimates, as mentioned also by Froidevaux
et al. (2019), we use a block bootstrap resampling method (Efron and Tibshirani, 1993), as done by Bourassa et
al. (2014), Mahieu et al. (2014), and others, in trend analyses of atmospheric composition. Basically, for every
fitted time series from MLS and the model, we analyze many (thousands of) resamplings of the fit residuals, with
year-long blocks of values replaced by values from randomly chosen years; (twice the) standard deviations in
these random distributions provide ($2\sigma$) uncertainty values. Such results are typically very similar to the 95%
confidence level (which would be arrived at by using the 2.5 and 97.5 percentile limits of the distributions). We
have found that such trend uncertainty calculations generally lead to significantly larger error bars than methods
that neglect the autocorrelation of the residuals, and even than some methods that include simple correction factors
for this autocorrelation (see more details in a later section).

## 3 Results

### 3.1 ClO

We first provide in Fig. 1 an overview of daytime ClO climatological values for January and July (averages

for 2005 through 2020) in the 50°S–50°N latitude region, and a comparison to the model results. As a consequence
of the photochemical balance between Cl and ClO radicals in the upper stratosphere, the largest ClO abundances
occur at pressure levels near 2 to 3 hPa; in the mid- to lower stratosphere, the availability of reactive chlorine is
limited by the conversion of ClO and $NO_2$ to $ClONO_2$. The observed ClO daytime distributions during January
and July are well reproduced by the model results (top and middle panels in Fig. 1, respectively), with ratios
between model and data between 0.9 and 1.1 for most latitudes at pressures less than 10 hPa (bottom panels in
Fig. 1); in this region, the systematic uncertainty estimates for MLS ClO are about 0.02 to 0.03 ppbv (see Livesey
et al., 2020), or of order 5–10%. Near 20–30 hPa, the model ClO values in the winter hemisphere mid- to high
latitudes are lower than observed by ~30%, although there is not much available ClO (in a climatological average
sense) in this region, and the systematic uncertainty estimates for MLS ClO are of order 0.1 ppbv, which can be
as much as 50–100%. Besides these features (and equally good model/data agreement during other months of the





year, not shown), we note that the model reproduces the seasonal changes in the peak ClO abundance patterns,
which are tied to other seasonal changes. Indeed, it has been shown in the past that seasonal and longer-term
variations in the $CH_4$ and $H_2O$ distributions play a primary role in the chlorine partitioning between upper
stratospheric HCl and ClO (see Solomon and Garcia, 1984; Siskind et al., 1998; Froidevaux et al., 2000).
Sample time series for the MLS ClO daytime data are shown in Fig. 2, along with the model series, and
regression fits (see Sect. 2) to both data and model series. Residual series are shown in the bottom panel of Fig. 2,
for the fits to MLS and to the model, and also for the model fit to MLS data, after taking out the average model
bias versus the data. In this latitude/pressure bin (35–40°N/2.2 hPa), there is a slight model underestimate of the
observed time series, but the modelled temporal decrease (reflected in the relevant fitted line) follows the slope
of the observed tendency fairly closely. The root mean square (rms) residual values for this panel, and in general,
are close to 5–7%, although the WACCM time series actually fit the MLS data better than the regression fits do,
as the rms residuals for (de-biased) WACCM versus MLS data are typically between 3 and 5%. These ClO results
are further quantified in Fig. 3, where we show excellent agreement between the modeled and observed trends
versus latitude at different pressures, in terms of the magnitude and morphology. These results demonstrate
statistically significant decreasing ClO trends of about –0.5 to –1%yr$^{-1}$ in the region between about 30 and 1 hPa
from 2005 to 2020, with very good agreement between the measurements and the WACCM6 simulations. Fig. 3
also shows that there is no significant difference between modelled and measured ClO trends, given the size of
the uncertainties (displayed in these plots as 2σ error bars), as obtained from the statistics of block bootstrap
resampling of the fitted residuals (see Sect. 2.3). This good agreement between modelled and measured ClO trends
can also be viewed in the pressure/latitude contour plots of Fig. 4; the trend differences (model minus data trends)
shown in the bottom panel are usually less than 0.1 to 0.2%yr$^{-1}$. In Fig. 5, we give the near-global (50°S to 50°N)
ClO profile trend results, based on our analyses of monthly zonal mean daytime profile time series for this region
as a whole. We obtain very similar trend values if we average results from separate latitude bins, or if we
deseasonalize time series from different (narrower) latitude bins prior to the regression. However, we feel it is
appropriate to apply the regression analysis to the whole 50°S to 50°N region to describe the resulting uncertainties
in these near-global trends in a consistent way, and (particularly) to compare overall ClO trends to those in other
species, as we do in a subsequent section. We see from Fig. 5 that measured near-global ClO trends are of order
–0.7 to –0.8%yr$^{-1}$ in the 15–1.5 hPa range, with values closer to –1%yr$^{-1}$ near 20 to 30 hPa. Model ClO trends are
typically slightly more negative than observed trends, with an average upper stratospheric value closer to
–0.9%yr$^{-1}$ (for pressures less than about 15 hPa). In summary, we find very good agreement in the derived ClO
trends between the model and the MLS data for 2005–2020, and the differences are not statistically significant.



### 267   3.2 HOCl

We now show results for HOCl, using the same approach as for ClO. The MLS HOCl offline product (see
Sect. 2.1) yields climatological fields displayed in Fig. 6 for January and July, over the 10 to 2 hPa region, where
the MLS HOCl data are deemed to be scientifically useful (see Livesey et al., 2020); this vertical range also holds
for the offline product. We observe peak HOCl January (daytime) values of about 160 pptv near 5 hPa at mid- to
high latitudes in the Southern Hemisphere, with slightly larger July peak values in the Northern Hemisphere (near
45°N). These patterns are also seen in the model HOCl (daytime) distributions, albeit with a shift to smaller
abundances; as seen from the model/MLS ratios in the bottom panels of Fig. 6, model HOCl values are typically
about 30% smaller than the mean measurements from MLS. This model-measurement difference is also seen in
the nighttime HOCl climatology, as shown in the supplementary material (Fig. S1). A small upward shift in the
altitude of peak nighttime HOCl abundances is seen in the MLS data, in comparison to the daytime case (Fig. 6),
as well as in the model values. Such a diurnal shift in the distribution of HOCl was also noted in the global satellite
measurements of HOCl made by the Michelson Interferometer for Passive Atmospheric Sounding (MIPAS)
aboard Envisat (von Clarmann et al., 2006; 2012). We note here that the MLS HOCl measurements have fairly
large systematic uncertainties ($2\sigma$ estimated systematic errors of 30–100%, see Livesey et al., 2020), which could
thus largely explain the model/data differences. We also note that slightly smoother profiles would be obtained
by applying the MLS averaging kernels to the model profiles, since the MLS HOCl vertical resolution is 5–6 km;
doing so would lead to an even larger model underestimate of the MLS HOCl profiles.
Another consideration to factor into the model uncertainties for HOCl has to do with the uncertainties in the
rate constant for HOCl formation ($k_{HO2+ClO}$). While the model used here conforms to the JPL Evaluation 18
(Burkholder et al., 2015) rate constant for this reaction, a more recent rate constant determination by Ward and
Rowley (2016) leads to significantly faster HOCl formation. Model simulations were performed to compare
annual mean HOCl abundances (50°S–50°N) based on these different choices of $k_{HO2+ClO}$, as shown in Fig. 7 (a);
the percent differences (in panel (b)) indicate that 25–45% larger HOCl abundances are obtained with the faster
rate constant, depending on altitude. The issue of a fairly poorly determined HOCl formation rate constant has
persisted for a number of years, affecting comparisons of balloon-borne HOCl profiles and model results
(Kovalenko et al., 2007), as well as analyses of MIPAS HOCl observations (von Clarmann et al., 2009; 2012).
Kovalenko et al. (2007) pointed out the need for a faster rate constant to improve agreement between modelled
and measured HOCl, such as the rate constant measured by Stimpfle et al. (1979), in comparison to the current
(at the time) value from the JPL Evaluation of Chemical Kinetics and Photochemical Data (Sander et al., 2006;



this position was supported by the MIPAS measurements of HOCl and other species over Antarctica (von
Clarmann et al., 2009). Using a temperature of 240 K, appropriate for the region of interest here, in previous
temperature-dependent laboratory studies leads to five different rate constant values that have oscillated over time.
Specifically, the values from Stimpfle et al. (1979), Nickolaisen et al. (2000), Knight et al. (2000), Hickson et al.
(2007), and Ward and Rowley (2016), respectively, yield 11.3, 10.3, 6.6, 8.6, and 12.5 (all in units of $10^{-12}$ cm$^3$
molecule$^{-1}$ s$^{-1}$), leading to an average of 9.7 with a ($1\sigma$) scatter of 2.1, or a range of about 3, if all five estimates
are included.  For comparison, the latest JPL Evaluation (Burkholder et al., 2019) gives an HOCl formation rate
constant of $8.7 \times 10^{-12}$ cm$^3$ molecule$^{-1}$ s$^{-1}$, although that particular report did not take into account the work from
Ward and Rowley (2016). However, making use of the Superconducting Submillimeter-Wave Limb-Emission
Sounder (SMILES) HOCl, ClO, and HO$_2$ data versus time of day, Kuribayashi et al. (2014) obtained a seemingly
well-constrained estimate of $k_{\mathrm{HO2+ClO}}$ for a limited temperature and pressure range ($7.75 \pm 0.25 \times 10^{-12}$ cm$^3$
molecule$^{-1}$ s$^{-1}$ at 245 K in the upper stratosphere). This leads to a value of ~$8.3 \times 10^{-12}$ at 240 K (as inferred using
an average temperature dependence), consistent with, but slightly smaller than, the latest evaluation's
recommendation mentioned above. To summarize, we find that the differences between MLS and model values
could well stem from a combination of uncertainties in both the MLS data and the model, and it is not possible to
definitively attribute the discrepancy to one or the other data set. This discussion does not include other uncertainty
sources (e.g., the photochemical loss rate of HOCl), as we believe that they are smaller in magnitude.

The MIPAS HOCl measurements were taken at about 10am/pm local time during 2002–2004; the SMILES

HOCl data cover the full diurnal cycle, but only for part of 2009–2010. The ACE-FTS solar occultation (i.e.,
sunrise/sunset) measurements have recently included retrievals of stratospheric HOCl profiles (up to about
38 km), as discussed by Bernath et al. (2021). The various satellite measurements of near-global HOCl
distributions are not easily compared, given their different local times and the non-negligible diurnal changes in
HOCl (see SPARC, 2017). Upper stratospheric peak HOCl values from ACE-FTS, MIPAS, Aura MLS, and
SMILES range from about 150 to 200 pptv, with MIPAS providing the largest values, as summarized by Bernath
et al. (2021). Khosravi et al. (2013) provided a more detailed intercomparison of HOCl measurements from
MIPAS, SMILES, and MLS in the upper stratosphere, with the help of model simulations of the diurnal cycle
(and ClO intercomparisons were also discussed). Good agreement was obtained, overall, versus the expected
HOCl diurnal variations, despite the noise in some of the data sets (with SMILES HOCl producing the least noisy
data). In SPARC (2017), HOCl monthly zonal mean distributions from MIPAS, SMILES, and MLS were
intercompared, albeit not for the same range of years (see also the recent update by Hegglin et al., 2021). Nighttime
values were used, as this time period exhibits somewhat smaller changes versus local time than the daytime data.





The MLS HOCl data were shown to be on the low side (by 20 to 30%) of both the MIPAS and SMILES results,
with the SMILES values lying between the MLS and MIPAS values; a low bias in MLS HOCl was also seen in
the comparisons presented by Khosravi et al. (2013). However, those studies used v3 HOCl data from the standard
MLS product. Mean differences between v3 HOCl and v5 HOCl are of order 5–10%, with the v5 data on the low
side of v3. More to the point, the offline HOCl retrievals yield larger values, by about 25%, than the monthly
zonal means from the standard v5 product, as can be seen from a comparison of Fig. 6 for the offline MLS HOCl
climatology versus Fig. S2 for the standard MLS HOCl product. The HOCl offline data values are thus about 20%
larger than the v3 MLS standard product values, so that much of the MLS low bias versus MIPAS and SMILES
is mitigated by using the offline MLS HOCl product. It follows from the above comments that the WACCM6
values will also significantly underestimate the HOCl abundances from MIPAS and SMILES. Based on the above
references discussing past satellite data intercomparisons for HOCl, the ($2\sigma$) systematic uncertainties for non-
MLS HOCl data sets are likely larger than 10–15%. The MLS v5 HOCl uncertainties are in the 40–80 pptv range
(see Livesey et al., 2020), or at least ~25% (and significantly more in the lower part of the upper stratosphere); it
is reasonable to expect that the offline MLS HOCl product will be affected by very similar systematic uncertainties
as the MLS standard product. In summary, we cannot expect much better agreement between the various HOCl
data sets than the (roughly) 20% level of agreement implied here.
Turning to the derived trends in HOCl, these will not be affected much (in units of $\%\,yr^{-1}$) by mean differences
between measured and modeled climatological values. As was done earlier for the ClO time series, we show
sample daytime HOCl time series, fits, and residuals in Fig. 8. We observe from such time series that, apart from
the absolute value difference between MLS and model HOCl, the measured seasonal cycle is well reproduced by
the model; less photochemical destruction of upper stratospheric HOCl during the winter months accounts for the
wintertime high values in the region shown (top panel). The residuals in this example (and in general) are larger,
by at least a factor of two, than those for ClO, and the correlation coefficients for the fits and for model versus
data are poorer, especially when comparing regression fits to the data and (de-biased) model fits to the data; the
poorer fits arise because the MLS HOCl data set is noisier (even for monthly zonal means) than is ClO. Thus, in
the case of HOCl, the regression fits to the model give the best results, in terms of correlation coefficients between
the regression fits to the MLS or model series, as well as for the de-biased model curves in comparison to the data,
and regarding root mean square residuals (as derived from data such as the curves in the bottom panel of Fig. 8).
The derived trends for HOCl are shown in Fig. S3 as a function of latitude, from 2.2 to 10 hPa. Many of the MLS-
derived trends at specific pressures and latitudes are not statistically different from a zero-trend value, while the
model-derived trends are typically negative (with values that are more negative than the measured trends) and


statistically different from zero. Figure 9 provides a summary of the results for MLS and model HOCl trends, with
day and night data shown separately, after multiple regression is applied to the averaged 50°S–50°N time series.
For MLS data between 3 and 7 hPa, we obtain statistically significant decreasing near-global HOCl trends, both
day and night. These results provide an unambiguous indication of decreasing upper stratospheric trends in HOCl,
given that negative trend center values occur at all retrieval levels. There is no statistically significant difference
between the nighttime and daytime results for either the MLS data or the model. The average model HOCl trend
($-0.6\%\,yr^{-1}$) is more negative than the average MLS result ($-0.4\%\,yr^{-1}$), although this is not a statistically significant
difference, given the ($2\sigma$) error bars shown in Fig. 9, and the fact that the MLS HOCl vertical resolution is about
6 km, so there are really only about 3 independent retrieval levels in the pressure range displayed in Fig. 9 (and
any error reduction for averaged results over all pressures would be by a factor of $\sqrt{3}$, or 1.7, at best). However,
the nighttime model and data trends at 2 hPa agree better than the daytime results, with the nighttime MLS trends
exhibiting a more homogeneous behavior versus pressure than the daytime MLS trends. This is likely caused by
the larger MLS signal for nighttime HOCl (see the climatological values in Fig. S1 versus the daytime values in
Fig. 6); the nighttime MLS trend errors are also smaller than the corresponding daytime errors.

We show in Fig. 10 a summary of the trend profiles for ClO and HOCl, both based on daytime results. We

mentioned above that the nighttime HOCl results agree well with those from daytime HOCl, and display better
agreement versus the model nighttime results at 2 hPa. For ClO, we have also checked that nighttime trends over
a limited pressure range (from 1.5 to 3.2 hPa) agree with the daytime trends (not shown), but nighttime ClO values
are typically much smaller than those during the day at pressures larger than 4 hPa, where we found that no robust
nighttime ClO trends can be obtained from the MLS data. Figure 10 demonstrates that both of these chlorine
species have decreased over much of the globe during the past 16 years, with the ClO trends being more negative
(by $\sim0.35\%\,yr^{-1}$) than the trends in HOCl, both in the model and the observational results. Limiting results to an
average over the uppermost stratosphere (between 2.2 and 6.8 hPa for both species), the (daytime) MLS-derived
near-global upper stratospheric trends are $-0.73 \pm 0.40\ \%\,yr^{-1}$ for ClO and $-0.39 \pm 0.35\ \%\,yr^{-1}$ for HOCl. The ($2\sigma$)
error bars here are the root mean square value applicable to this vertical range, with no reduction in error bars for
the broader region; we would rather use a somewhat more conservative uncertainty than one that is too
"optimistic" (such as an error reduction by a factor of two for ClO, which assumes uncorrelated errors between
pressure levels). The corresponding model trends for this vertical range are $-0.85 \pm 0.45\ \%\,yr^{-1}$ for ClO and $-0.64$
$\pm 0.37\ \%\,yr^{-1}$ for HOCl. Even if the HOCl trends are not significantly different from the ClO trends at any given
level, when averaged, these differences do become more significant.





## 4 Discussion


We now review our estimated trends in the context of past results, and we discuss potential reasons for different
trends in various chlorine species in the upper stratosphere, including the slower decrease in upper stratospheric
HOCl in comparison to the ClO decrease. As a reminder of the relative importance of the main inorganic chlorine
species in the upper stratosphere, we display in Fig. 11 the percent contribution to total inorganic chlorine ($Cl_y$)
over the 10 to 1 hPa range, based on the climatological (daytime) model results over 50°S–50°N for the time
period analyzed here. The Cly abundance includes all species contributions from HCl, ClO, HOCl, and $ClONO_2$,
which are shown in the plot, as well as very minor contributions from Cl, $Cl_2$, $Cl_2O_2$, OClO, and BrCl. The "Sum"
curve shown on the right side of this figure is just the sum from the four main species whose contributions are
plotted; this does not quite equal 100% because of the very small (daytime) relative contributions from the latter
five species. HCl is clearly the dominant reservoir in the upper stratosphere, as it makes up about 80 to 95% of
total inorganic chlorine in this region (see also Froidevaux et al., 2006), while ClO makes up about 5 to 15% of
the total, with minor contributions from $ClONO_2$ and HOCl, both at the few percent level for most of this region.
While published trends in chlorine species can be compared, there will always be some differences in the
results, given the different measurement locations, coverage, and time periods being considered. We note that the
surface maximum in total chlorine was reached in 1992–1993; following the fast initial decrease in methyl
chloroform ($CH_3CCl_3$), tropospheric chlorine declined at a slower rate (O'Doherty et al., 2004). There is also
evidence for slightly slower decreases in the ACE-FTS upper stratospheric HCl time series after about 2010
(Bernath and Fernando, 2018; Bernath et al., 2020), in comparison to the rate of decline over the 2004–2010
period. In terms of the MLS ClO results discussed here, the upper stratospheric trend (for 2005–2020) of
–0.73 ± 0.40 %yr⁻¹ can be compared to other estimated trends in upper stratospheric ClO. Jones et al. (2011)
reported upper stratospheric ClO trends of –0.7 ± 0.8 %yr⁻¹ for 2001 through 2008, based on a combination of
Odin Sub-Millimetre Radiometer (SMR) and Aura MLS data over the tropics; the estimated uncertainty in this
satellite-based ClO trend is quite large, but the trend estimate is consistent with our result covering a longer time
period. Solomon et al. (2006) displayed the rise and decline of upper stratospheric ClO abundances in the 1982 to
2004 time period, based on microwave ground-based profile data from Hawaii. However, the fairly large ClO
trend (–1.5%yr⁻¹) initially obtained by these authors for 1995–2004 was superseded by analyses of an improved
data set over a longer time period using a new methodology (Connor et al., 2013), which led to a ClO trend
estimate (at about 4 hPa) of –0.65 ± 0.15 (2σ) %yr⁻¹ over the 1995–2012 period. Thus, we find good consistency



between our MLS results and previous trend estimates for ClO, especially given the differences in measurement
coverage and time periods considered.

For the HOCl trends, we are aware of only one prior result, a recent trend estimate based on ACE-FTS HOCl

data by Bernath et al. (2021), who quote a marginally significant trend of $-0.23 \pm 0.22$ ($2\sigma$) pptv yr$^{-1}$, which we
translate to about $-0.19 \pm 0.18$ %yr$^{-1}$, given mean HOCl abundances (of 124 pptv) from their analysis of ACE-
FTS data at 30–39 km and 60°S–60°N from 2004–2020. This can be compared to our near-global MLS HOCl
trend estimate of $-0.39 \pm 0.35$ %yr$^{-1}$ for a very similar time period; while these two estimates agree within the
fairly large uncertainty estimates, the MLS mean trend value represents twice as rapid a decrease as the mean
ACE-FTS trend result. At this time, the cause of these differences is not known, although these measurements are
among the more difficult for both instruments, and the two sampling patterns are quite different. We note that the
model upper stratospheric HOCl trend is faster (at $-0.64 \pm 0.37$ %yr$^{-1}$) than the MLS-derived trend, and even
faster in comparison to the ACE-FTS result.

We now turn to some additional model results as well as other relevant measurements from MLS and ACE-

FTS, to discuss upper stratospheric trends in chlorine and related species in a broader context. Figure 12 shows
the derived average trends in various upper stratospheric chlorine species based on our regression analyses of
measured and modeled time series for monthly zonal means from 50°S to 50°N. The near-global upper
stratospheric trend values in Fig. 12 are obtained from trends like those in Fig. 10 for MLS ClO and HOCl, but
averaged from 6.8 to 2.2 hPa. Error bars represent typical $2\sigma$ estimates, calculated from the root mean square of
the $2\sigma$ estimates for pressures in the 6.8 to 2.2 hPa range; we prefer to use this more conservative error rather than
the standard error in the mean, which will typically be an underestimate, since errors from different pressure levels
are not completely uncorrelated. As mentioned earlier, no useful MLS-based estimate of HCl trends in the upper
stratosphere could be obtained after the related MLS hardware degradation in early 2006. MLS HCl measurements
are still scientifically useful in the lower stratosphere, even for trends (see the related model/data analysis by
Froidevaux et al., 2019), and certainly they accurately capture the larger seasonal, interannual, and winter polar
vortex HCl variations. To derive the trends based on ACE-FTS data shown in Fig. 12, we have used seasonally
averaged time series of v4.1 measurements, a methodology used in previous investigations of ACE-FTS trends to
lessen the impacts of that instrument's sampling patterns (e.g., see Bernath and Fernando, 2018). We have applied
a simple linear fit to the deseasonalized anomalies from ACE-FTS seasonal means (from 50°S to 50°N), thus
using the same type of analysis as in the latter reference. In this approach, the auto-correlation of the residuals is
taken into account by following the methodology described by Tiao et al. (1990) and Weatherhead et al. (1998);





the auto-correlation is assumed to follow a first-order autoregressive model, and the trend error bars are multiplied
by a factor that depends on the autoregressive coefficient. We also point out that it would be more complicated to
apply the MLR approach used for the MLS and model time series to the ACE-FTS seasonal data, as the MLR
method we have used is based on monthly proxy values. A careful intercomparison of different approaches to
estimate error bars in various trends analyses is beyond the scope of this paper, although such an intercomparison
would be helpful.
We see in Fig. 12 (as was shown in Fig. 10) that the MLS ClO trend is more negative than the MLS HOCl
trend; this is also true for the model results in Fig. 12, and the model ClO trend is also more negative than the
model $Cl_y$ and HCl trends (with respective values of $-0.66 \pm 0.30$ %yr$^{-1}$ and $-0.64$ % $\pm$ 0.30 %yr$^{-1}$ ($2\sigma$)). The
faster ClO decrease (versus $Cl_y$ or HCl) seen in Fig. 12 is tied to the dependence of ClO on other species. More
specifically, the ClO abundance ([ClO]) is roughly proportional to [HCl] $[H_2O]^{1/2}$ / $[CH_4]$ (see Froidevaux et al.,
2000). The model and observed trends in both $H_2O$ and $CH_4$ agree well (see the bottom portion of Fig. 12). Here,
we have averaged all ACE-FTS (50°S–50°N) trends between 33 and 43 km, based on all sunrise and sunset
profiles combined. The MLS v5 $H_2O$ trend of $0.13 \pm 0.15$ ($2\sigma$) %yr$^{-1}$ is close to the trend we obtain from ACE-
FTS data, at $0.18 \pm 0.15$ %yr$^{-1}$ (which is in reasonable agreement with the near-global mid-stratospheric $H_2O$ trend
of 0.24 %yr$^{-1}$ provided in the broad overview of ACE-FTS trends by Bernath et al., 2020). Although the MLS v4
$H_2O$ data suffered from a drift that led to trends that were too large, this drift has been largely mitigated in the v5
$H_2O$ data used here (Livesey et al., 2021). The measured trend in $CH_4$, also obtained from ACE-FTS data, as well
as the model $CH_4$ trend (in very good agreement with the ACE-FTS trend), are significantly larger than the trends
in $H_2O$; more $CH_4$ will thus lead, in time, to less chlorine in the form of ClO, which means a faster rate of decrease
for ClO. The photochemical balance for HOCl, on the other hand, leads to [HOCl] being roughly proportional to
$k_{HO2+ClO}$ [ClO] $[HO_2]$ / $(J_{HOCl} + k_{HOCl+OH}$ [OH] $+ k_{HOCl+O}$ [O]), where $J_{HOCl}$ is the photodissociation rate constant for
HOCl, and the rate constants indicate which HOCl production or destruction reaction we are referring to. In the
mid- to upper stratosphere, the $J$ term clearly dominates (e.g., see Chance et al., 1989, and also, based on our
diagnostics for the WACCM run used here), and we would thus expect the trend in HOCl to be less negative than
the trend in ClO, given that the $HO_2$ trend is (slightly) positive (per Fig. 12). The MLS-derived trend for $HO_2$
comes from our analysis of the offline MLS $HO_2$ product (see Millán et al., 2015). As recommended for this
product, we performed our trend analysis using day minus night differences, that is, we constructed such monthly
zonal means from the set of day and night daily zonal means; the model and data $HO_2$ trends agree within the
error bars, although the MLS error bar is quite large. The model OH trend also points to a slight positive trend,
which likely stems from the increasing trends in $H_2O$. Algebraically, a percent change in HOCl will be driven by





the percent change in ClO added to the percent change in $HO_2$, so that the decreasing trend in HOCl is slowed,
relative to the ClO trend, by the increasing trend in $HO_2$. Using the modelled $HO_2$ trend in Fig. 12 (~0.2%yr$^{-1}$),
which is consistent with the observed $HO_2$ trend, one could expect the HOCl trend to lie ~0.2% closer to zero than
the ClO trend; this is consistent (within the error bars) with both the modelled and measured trend differences
between HOCl and ClO (these differences are ~0.2% and 0.3%, respectively, for the model and for the
measurements).

The $ClONO_2$ trends shown in Fig. 12 are less negative than the ClO trends; this likely stems from the slightly

positive trends in $NO_2$, which can mitigate the extent of the decrease in $ClONO_2$ (formed from ClO and $NO_2$). We
also note that the differences between the model and ACE-FTS HCl trends are somewhat larger than those between
the model and MLS ClO, although the error bars in Fig. 12 indicate that none of these differences are statistically
significant. It has been shown that the better sampling from emission-type measurements can provide more
reliable trend estimates than in the case of sparser sampling (e.g., from occultation-type data; see Millán et al.,
2016). We expect that sampling differences between ACE-FTS and MLS (or the model) contribute part of the
trend differences versus MLS (or the model). In this regard, error bars in the ACE-FTS trends are likely to be
smaller than the errors that would be obtained from a more fully sampled dataset with less data averaging (and
thus, with more spatio-temporal variability).

While this is less pertinent to the chlorine species trends, we find it interesting that the $N_2O$ trends in Fig. 12

appear to be much larger than the trends in NO and $NO_2$, two radicals that are the products of $N_2O$ destruction in
the upper stratosphere; MLS, ACE-FTS, and the model results all point to upper stratospheric trends slightly larger
than 1%yr$^{-1}$, albeit with comparable $2\sigma$ uncertainties. Some of this difference might be caused by the strong
latitude dependence of the $N_2O$ trends, coupled with large trend uncertainties in a region with rapidly decreasing
abundances with height; the $N_2O$ trends from ACE-FTS at lower altitudes yield small positive values that are
more consistent with the $NO_x$ trends shown here, and also with tropospheric $N_2O$ trends (see also Bernath et al.,
2020). We note also that the MLS $N_2O$ trends likely constitute lower limits, given that there are some unmitigated
negative drifts in the version 5 MLS $N_2O$ time series in the lower stratosphere, even after the improvements versus
the v4 data (Livesey et al., 2021). Finally, there are also temperature-related effects that could potentially modify
the partitioning of chlorine species over the long-term. However, since the average upper stratospheric
temperature decrease over the past 16 years is less than 1K (e.g., Steiner et al., 2020), the temperature dependence
issue for this time period should not lead to a significant perturbation of chlorine species trends and chlorine
partitioning in this region. For the ClO or HOCl photochemical balance in particular, the strongest temperature-
dependence (by far) is from the Cl + $CH_4$ reaction, but even this would lead to a fairly small (15-30%) perturbation



(for the cooling rate implied above) in comparison to the impact of the $CH_4$ trend, or versus the trends in the
chlorine species themselves.
We have provided above a few arguments that can help explain some of the differences in upper stratospheric
chlorine species trends summarized in Fig. 12. The full chemistry climate model takes all the (modelled) factors
into account, both regarding photochemical balance issues and any underlying dynamical factors, such as
variations and trends in long-lived tracers that can also impact shorter-lived species.

## 516   4 Conclusions

We have analyzed Aura MLS monthly zonal mean time series of ClO and HOCl between 50ºS and 50ºN to
estimate upper stratospheric trends in these chlorine species from 2005 through 2020. We compare these
observations to those from a state-of-the-art chemistry climate model, WACCM6, run under the specified
dynamics configuration, with MERRA-2 meteorological constraints, and sampled for the same time period; in
addition, the model sampling follows the MLS coverage in space and local time. We use version 5 MLS ClO
zonal mean (Level 3) daytime profiles (associated with solar zenith angles less than 90º) and, for comparison,
similarly binned daytime ClO model profiles. For MLS HOCl, we use the version 5 offline product derived from
daily zonal mean radiances (in 10º latitude bins) rather than averaged Level 2 profiles; MLS HOCl is scientifically
useful between 10 and 2 hPa, and HOCl monthly zonal means are separated into day and night averages (solar
zenith angles greater than 100º for night conditions), for comparison to similarly binned WACCM6 HOCl profiles.
We find good agreement (mostly within about 10%) between the climatological MLS daytime ClO
distributions and the corresponding model ClO climatology for 2005–2020. The model HOCl climatology,
however, underestimates the MLS HOCl climatology by about 30% (for both daytime and nighttime). This
discrepancy could well be caused by a combination of fairly large systematic uncertainties in both the model-
assumed rate constant for the formation of HOCl and the MLS HOCl retrievals themselves, although we note that
these model results would likely also underestimate other satellite measurements of HOCl. The model daytime
ClO trends versus latitude and pressure agree well with those from MLS ClO. MLS-derived near-global upper
stratospheric daytime trends between 7 and 2 hPa are $-0.73 \pm 0.40$ %yr$^{-1}$ for ClO and $-0.39 \pm 0.35$ %yr$^{-1}$ for HOCl,
with $2\sigma$ uncertainty estimates used here. The corresponding near-global upper stratospheric model trends are
$-0.85 \pm 0.45$ %yr$^{-1}$ for ClO and $-0.64 \pm 0.37$ %yr$^{-1}$ for HOCl. Both data and model results point to a slower trend
for HOCl than for ClO. The MLS trends for ClO are generally consistent with past estimates of upper stratospheric
ClO trends, based on a combination of Odin/SMR and MLS data from 2001 to 2008 (Jones et al., 2011), and based





on ground-based microwave results from Hawaii for 1995–2012 (Connor et al., 2013). The MLS HOCl trend
represents a faster rate of change (by about a factor of two) than the marginally significant trend (–0.19 ± 0.18
(2σ) %yr$^{-1}$) that Bernath et al. (2021) obtained from a recent analysis of ACE-FTS HOCl measurements from
2004 to 2020.
Our general overview (Fig. 12) shows decreasing near-global trends for all the measured upper stratospheric
chlorine species. Differences can arise as a result of the impact of trends in other gases that can affect the slowly
varying photochemical equilibrium for different species in this region. Notably, observed and modeled positive
trends in CH$_4$ will tend to steepen the decrease of active chlorine (ClO values) in comparison to trends in HCl or
Cl$_y$. Regarding trends in HOCl, positive trends in HO$_2$ can lead to a faster rate of formation for HOCl as a function
of time, which partially offsets the impact of decreases in ClO (also involved in HOCl production).
Lastly, the decreasing trends in upper stratospheric ClO and HOCl that are arrived at in this work provide
additional confirmation of the effectiveness of the Montreal Protocol and its amendments, which have led to the
early stages of an expected long-term ozone recovery from the effects of ozone-depleting substances (see WMO,
2018). Indeed, the known decreases in surface chlorine since the early 1990s, which are faithfully included in the
model results, have played a major role in the decreasing trends of ClO and HOCl over the 2005–2020 time period.




*Data availability.* The link http://disc.sci.gsfc.nasa.gov/Aura/data-holdings/MLS provides public access to Aura
MLS data used here; for the offline MLS HOCl product, the data are available upon request to Luis Millán
(luis.f.millan@jpl.nasa.gov). For the availability of ACE-FTS 4.1 data, see http://www.ace.uwaterloo.ca/data.php
(registration required at https://database.scisat.ca/l2signup.php). For solar flux data, the site
ftp://ftp.seismo.nrcan.gc.ca/spaceweather/solar_flux/monthly_averages/solflux_monthly_average.txt was used to
obtain monthly means of the Canadian F10.7 solar flux measurements; these series (see
http://www.spaceweather.gc.ca) were included in our regression fits. MERRA-2 data can be obtained from NASA
at https://gmao.gsfc.nasa.gov/reanalysis/MERRA-2/data_access/. Model results shown in this paper are available
online at:
https://urldefense.us/v3/__https://acomstaff.acom.ucar.edu/dkin/ACP_Froidevaux_2021/__;!!PvBDto6Hs4WbV
uu7!Yk6MAjKksie5II_GsOQzm_FmoXFSt_0ExPIxMEA6hEUdqF6I4S72h5M3WBjsxoKwX1vZRj9wgz6b$.





*Author Contributions.* LF prepared this manuscript with contributions from all co-authors. DEK provided inputs
for running the necessary model runs, as well as properly averaged and formatted outputs from the model, along
with various contributions to the main text and Figures. MLS provided assistance in the validation and generation
of the MLS ClO data, along with comments on the manuscript. LFM provided the MLS HOCl offline products
and related expertise, along with comments on the manuscript. NJL provided leadership for MLS overall, along
with comments regarding the manuscript. WGR provided leadership for the MLS forward model and data
retrievals, along with measurement science expertise and related manuscript comments. CGB provided assistance
towards obtaining the model runs, as well as comments regarding the model description. JJO provided assistance
regarding the available laboratory data on the HOCl formation rate constant and its related uncertainties, along
with manuscript comments. RAF provided properly formatted and averaged MLS data sets for the various species
analyzed in this work.

*Competing Interests.* The authors declare that they have no conflict of interest.

*Acknowledgments.* We are thankful to the whole MLS team (past and present) for their contributions over the
years to the MLS instrument, data, processing, and database management; all this has contributed to making this
research work possible. We also gratefully acknowledge the work of the whole ACE-FTS team in producing and
sharing their updated data sets, as these were used here as part of the discussion and comparisons. F10.7 data
collection and dissemination are supported by the National Research Council of Canada, with the participation of
Natural Resources Canada and support by the Canadian Space Agency. D.E.K. was funded in part by NASA grant
(80NSSC20K0926). WACCM is a component of the CESM, supported by the National Science
Foundation (NSF). We would like to acknowledge high-performance computing support from Cheyenne
(doi:10.5065/D6RX99HX) provided by NCAR's Computational and Information Systems Laboratory, sponsored
by the NSF. Work at the Jet Propulsion Laboratory, California Institute of Technology, was performed under
contract with the National Aeronautics and Space Administration (80NM0018D0004). Copyright 2021. All rights
reserved.




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

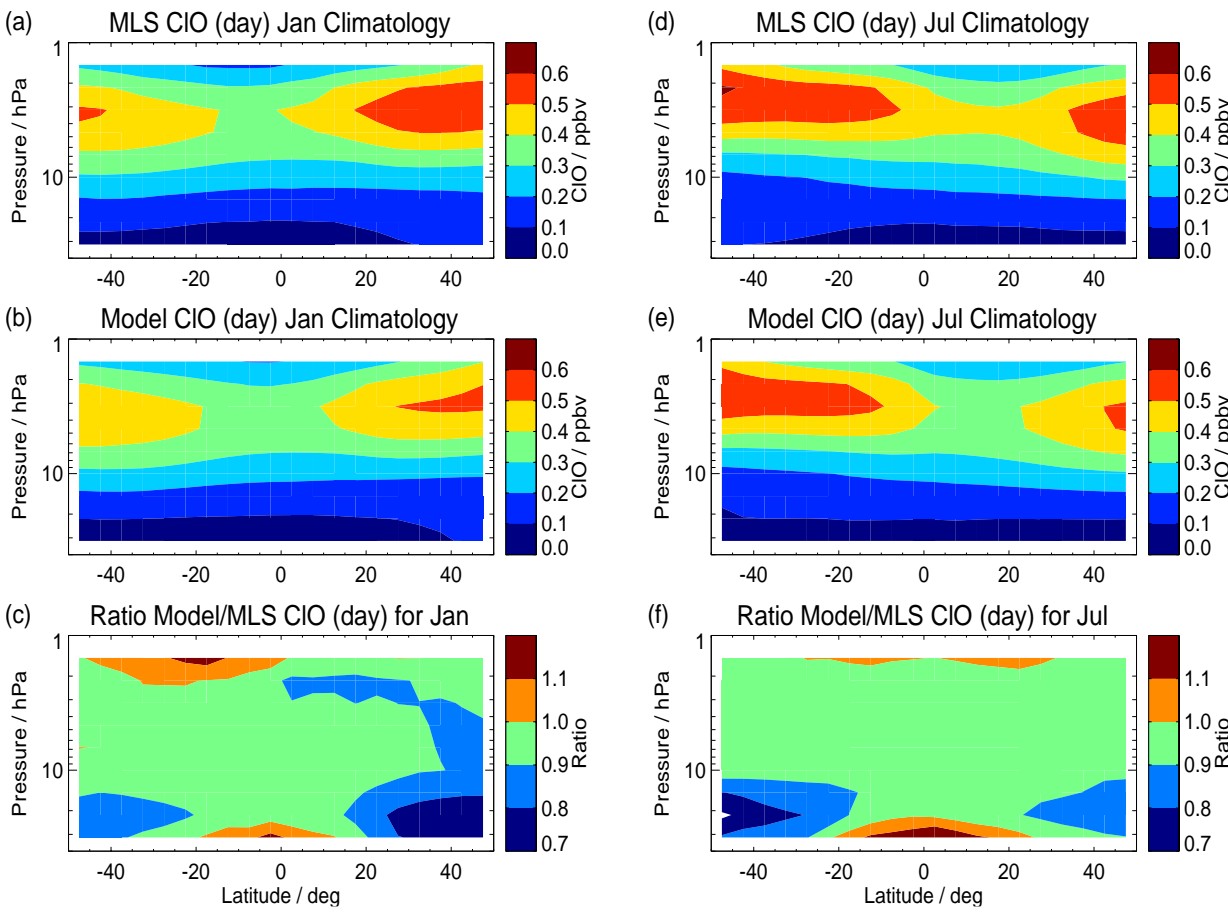


**Figure 1.** Climatological mean fields (over 2005 through 2020 for ClO (daytime data) from MLS and model results between
50°S and 50°N and 32 to 1.5 hPa. Daytime averages (observed and simulated values) are based on values with solar zenith
angles less than 90° only. (a) and (b) show the January MLS and model climatologies, respectively, while (c) gives the ratio
(model values divided by MLS values) for that month; (d), (e), and (f) are the same as (a), (b), and (c), respectively, but for
July instead of January. The model daily values (throughout this work) were sampled to provide the closest match in space
and time to the MLS daily Level 2 data; model results were then binned in latitude and averaged over each month, and
interpolated to the MLS pressure grid, in order to best match the averaging process of MLS monthly zonal mean data.

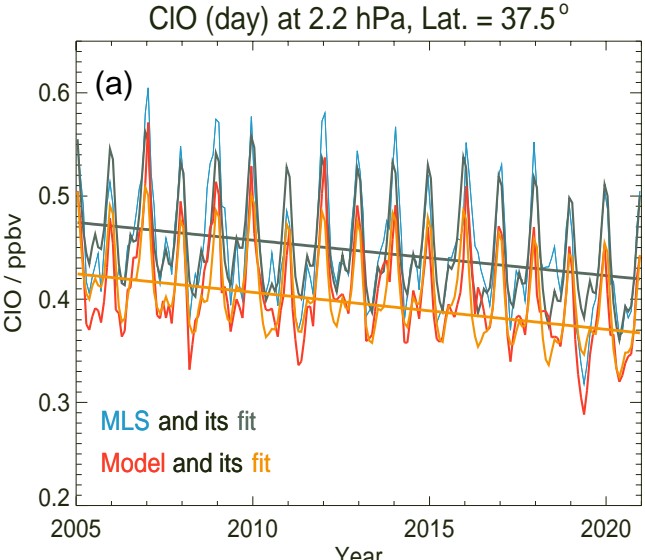

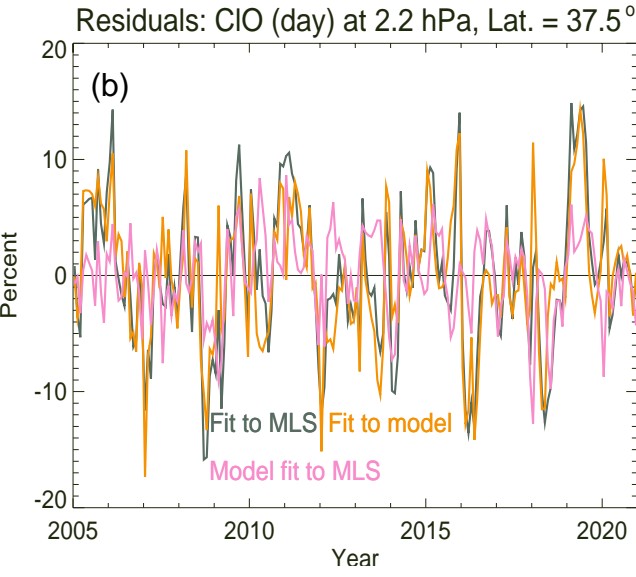

**Figure 2.** (a) Examples of MLS and model ClO (day) monthly zonal mean time series (2005 through 2020) for the 35°N–
40°N latitude bin at 2.2 hPa. The MLS data (blue) are fitted by a regression model (grey), and the model series (red) is fitted
by the same type of regression model (orange). The grey and orange lines are the linear components of the corresponding fits
to the MLS and model curves, respectively. (b) Residuals, with the fit to MLS (minus MLS) in grey, the fit to the model (minus
the model) in orange, and the de-biased model fit to MLS (minus MLS) in pink.



**Figure 3.** Linear trends in upper stratospheric ClO (2005 through 2020) at different pressure levels versus latitude, as obtained
from multiple regression analyses applied to monthly zonal mean daytime series from MLS (blue) and the model (red). Error
bars depict the uncertainties (2σ) for these trend results, based on block bootstrap analyses of the monthly residual series from
the fits to the MLS and model series.



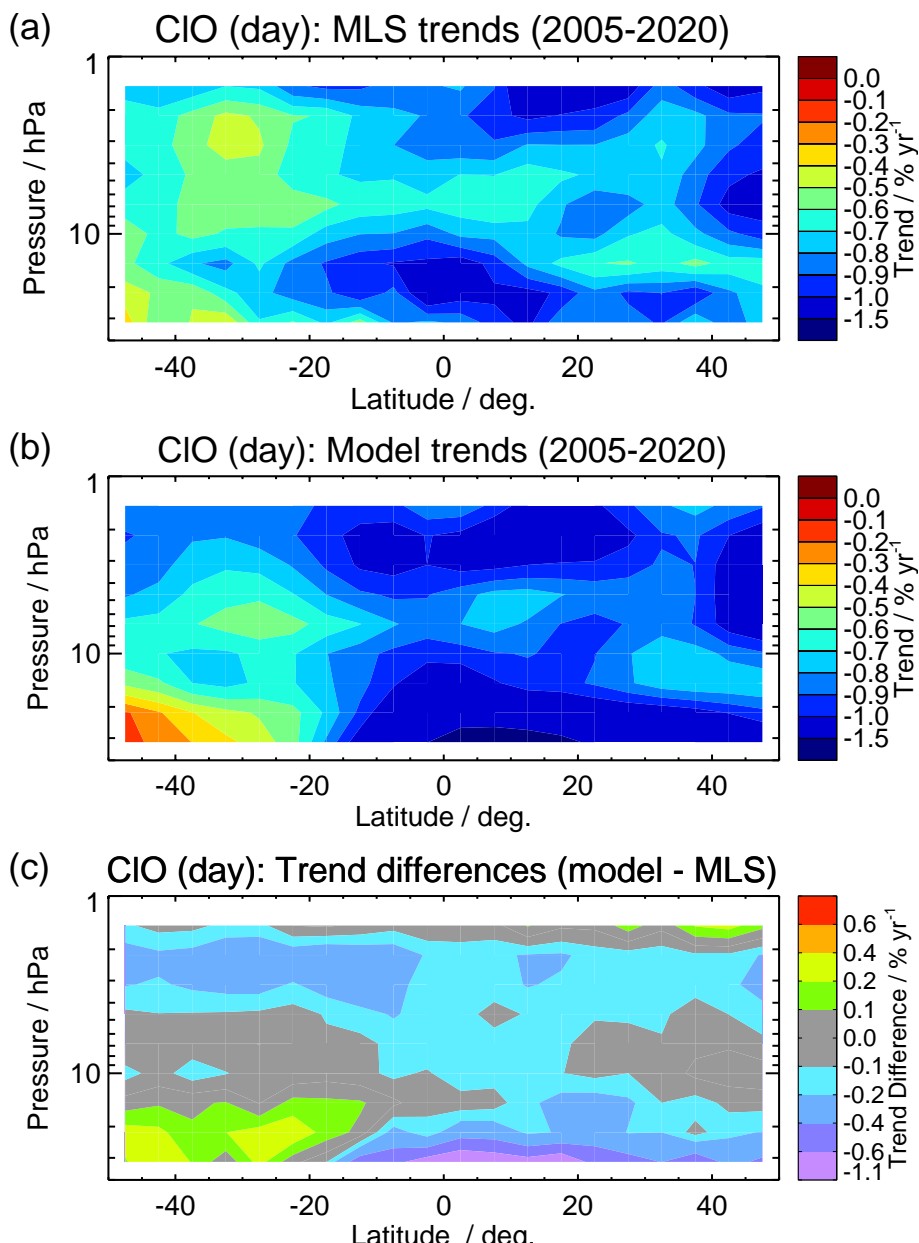

**Figure 4.** Contour plots of ClO (day) trends (%yr$^{-1}$) for the period 2005 through 2020 from (a) MLS, and (b) model, with (c)
showing the differences (%yr$^{-1}$) in these trends (model – MLS).

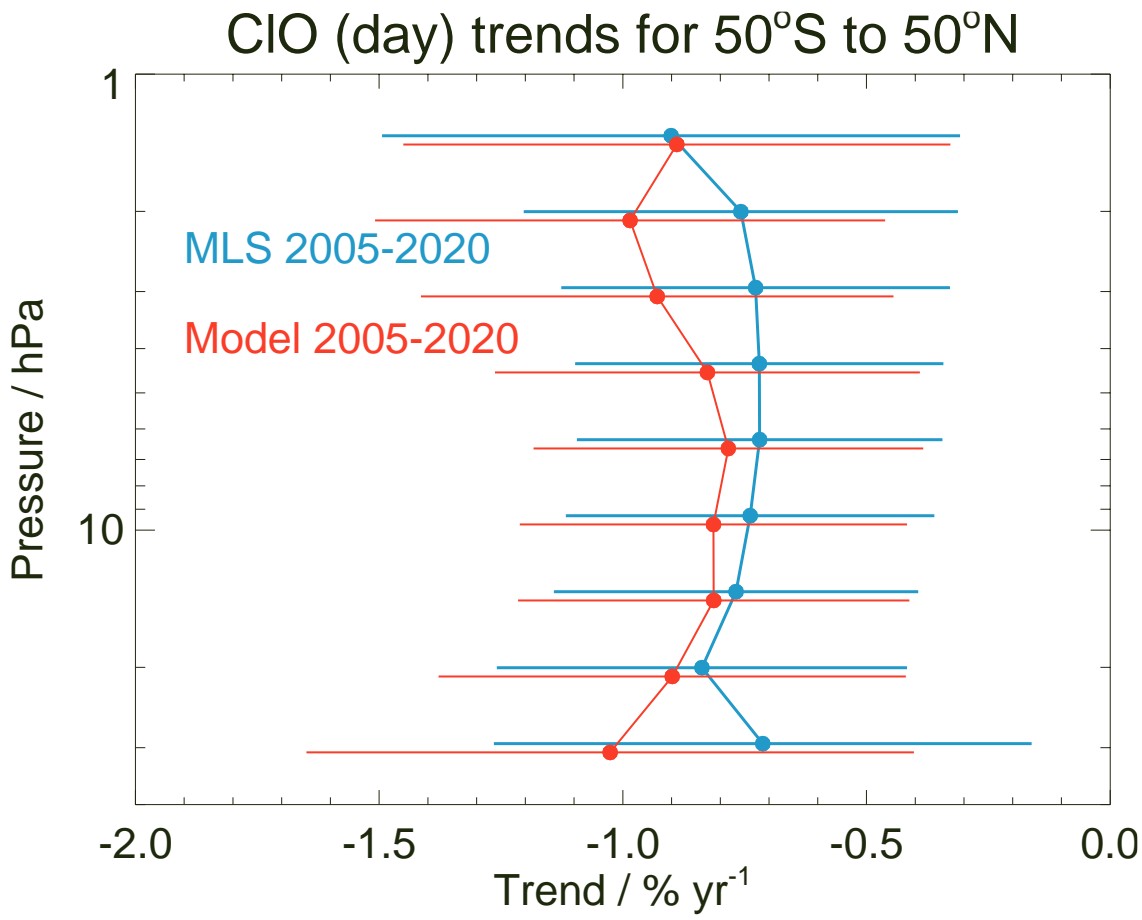

**Figure 5.** Trends in ClO (daytime values) over 2005 through 2020 from MLS (blue) and model (red) for the 50°S to 50°N
latitude range. Error bars depict the uncertainties (2σ) for these trend results, based on block bootstrap analyses of the monthly
residual series from the fits to the MLS and model time series.



**Figure 6.** Same as Figure 1, except for climatological (2005–2020) HOCl daytime values from MLS and the model (see text for more details); the vertical range for useful MLS HOCl data (and for related trend analyses) is 10 to 2.2 hPa.



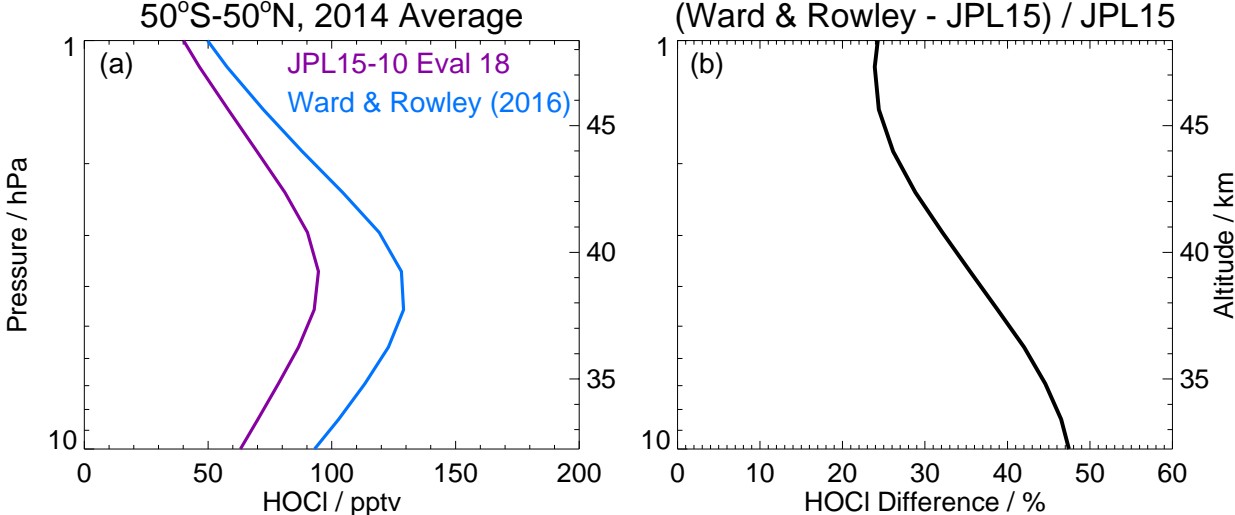

**Figure 7.** (a) Sensitivity of average (for 2014, 50°S to 50°N) model upper stratospheric HOCl profile (pptv) to the choice of
rate constant for the HOCl formation reaction between $HO_2$ and ClO. The JPL 15-10 Evaluation 18 rate constant choice gives
the purple average profile, whereas the larger rate constant derived by Ward and Rowley (2016) leads to the blue average
profile. (b) The percent difference (increase) between the two curves in panel (a) (blue minus purple).

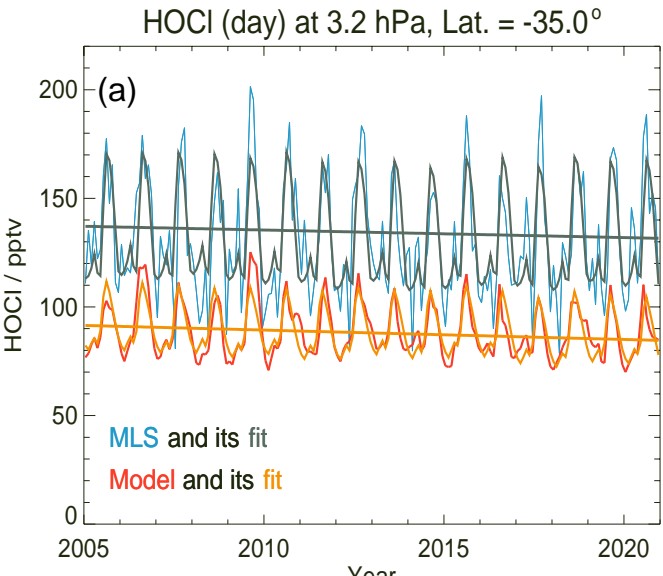

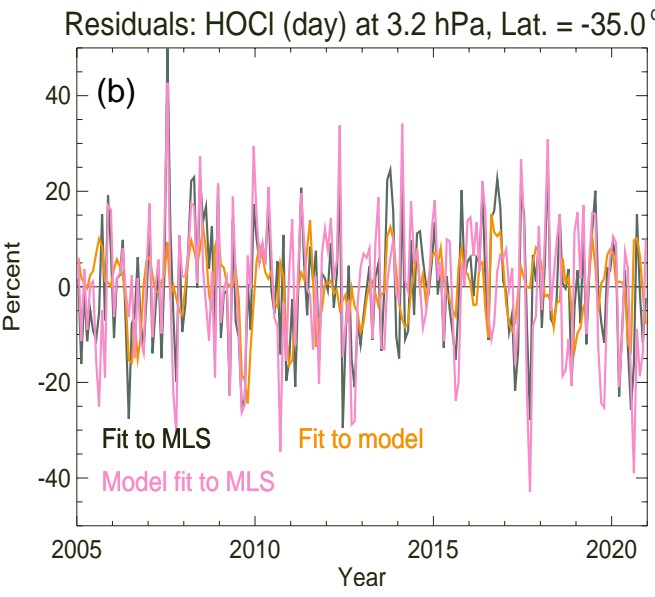

**Figure. 8.** Same as Fig. 2, except for an example of (a) HOCl time series and regression fits and (b) residuals for 3.2 hPa and
the 30°S to 40°S latitude bin.



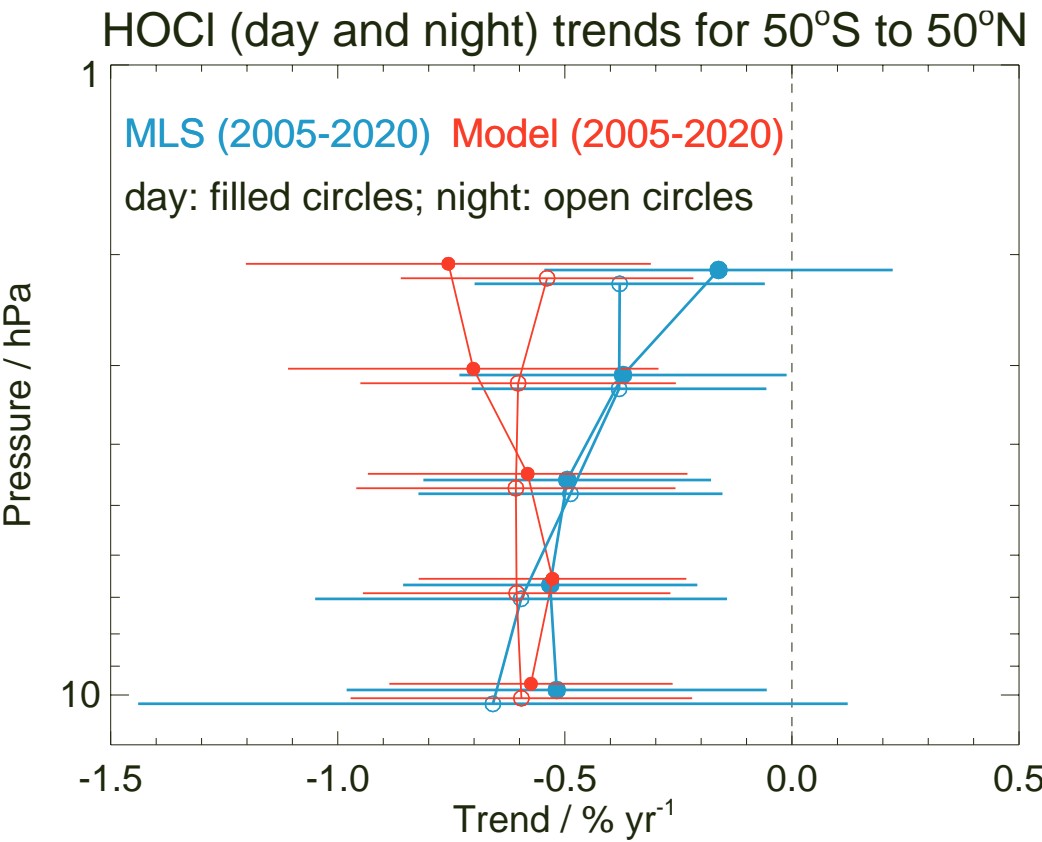

**Figure 9.** Same as Fig. 5, except for trend results for HOCl from both day (filled circles) and night (open circles) time series
analyses between 10 and 2.2 hPa.



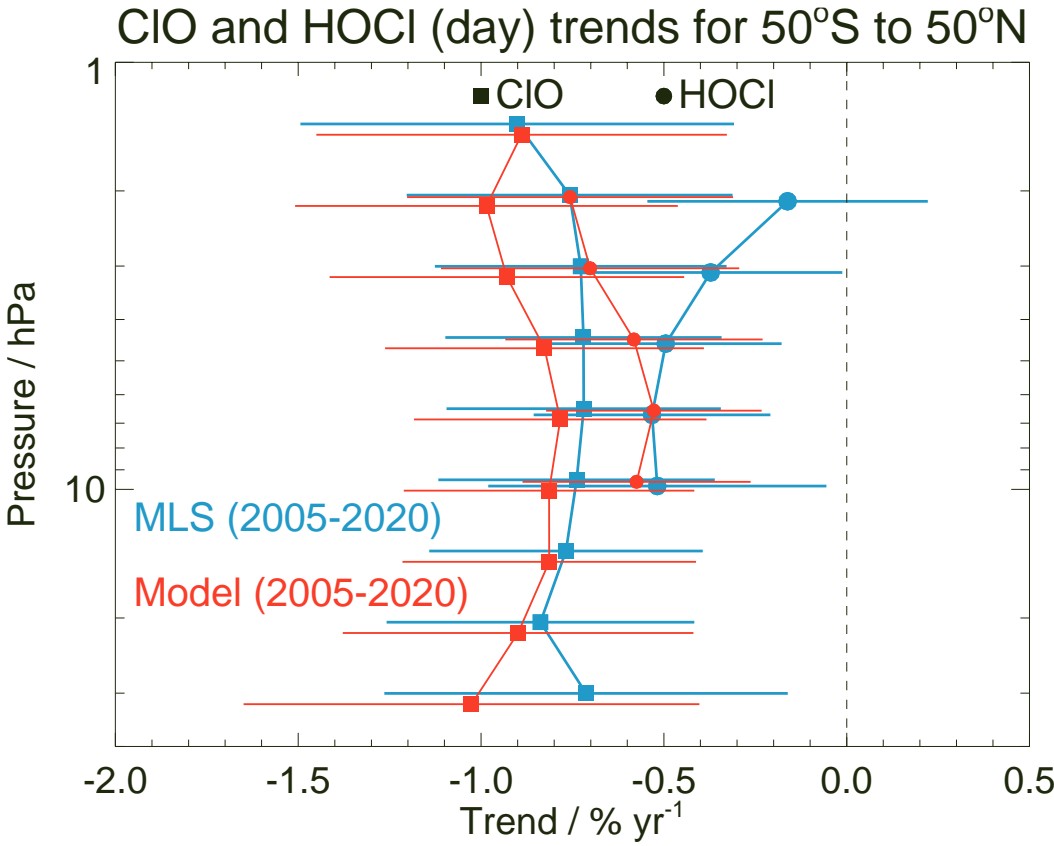

**Figure 10.** Derived upper stratospheric trends in ClO (filled squares) and HOCl (filled circles) based on regression fits to
daytime monthly zonal mean time series for both species, for 50°S to 50°N averages from 2005 through 2020; MLS results are
in blue and model results in red.



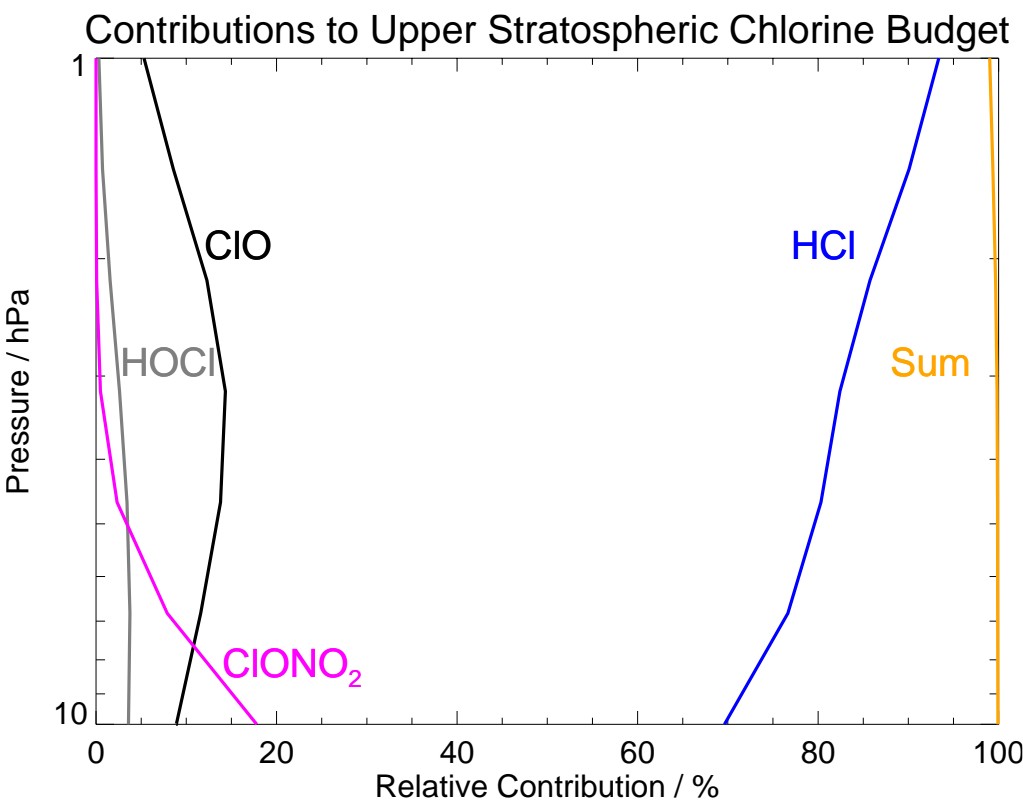

**Figure 11.** Percent contributions of various species (daytime HCl, ClO, HOCl, and ClONO$_2$) to the upper stratospheric chlorine
budget between 10 and 1 hPa, based on climatological (16-yr) daytime model results in the 50ºS to 50ºN latitude range. The
sum of these contributions is shown in orange; there are also very small contributions in this pressure range from other species
(Cl, Cl$_2$, Cl$_2$O$_2$, OClO, BrCl, which are not represented here).

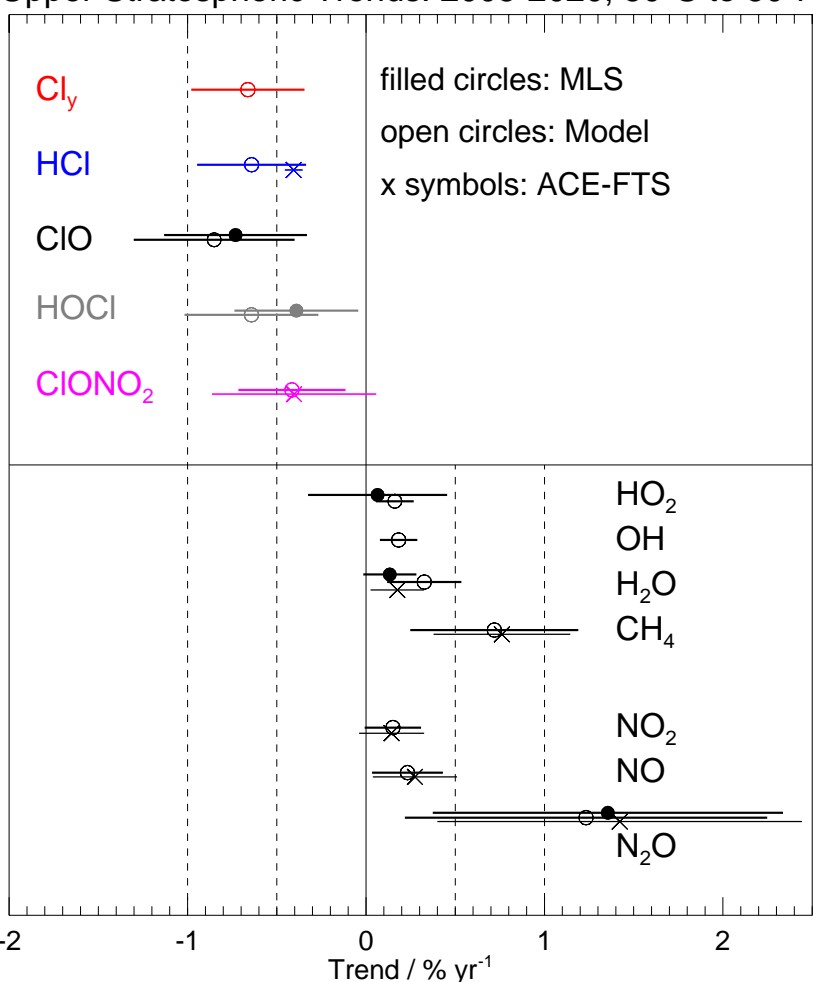

**Figure 12.** Upper stratospheric trends in various species from 6.8 to 2.2 hPa for 50°S to 50°N, based on linear trends obtained
from the regression fits to daytime time series of MLS data (filled circles) and/or model series (open circles); x symbols are
from our analysis of (50°S to 50°N) ACE-FTS version 4.1 data over the 33 to 43 km range (see text). Error bars represent
uncertainties (2σ), derived as described in the text.