# Peer review of "Upper stratospheric ClO and HOCl trends (2005–2020): Aura Microwave Limb Sounder and model results"

_Atmospheric Chemistry and Physics, 2021_

## Author Response (AR2)

**Comment on acp-2021-880**
**Anonymous Referee #1**

*We thank the referee for the thoughtful comments and suggestions; we address these (in italics) in the text below under* **"Answer"**, *for each point made. Our revised document will clearly include the changes we point to in detail below.*

**Referee comment on "Upper stratospheric ClO and HOCl trends (2005–2020): Aura Microwave Limb Sounder and model results" by Lucien Froidevaux et al., Atmos. Chem. Phys. Discuss., https://doi.org/10.5194/acp-2021-880-RC1, 2021**

This is an excellent paper, using results both from MLS measurements and the WACCM model to obtain trend estimates of ClO and HOCl over the ~15 years for which MLS has been making measurements. With the exception of requiring a better explanation for a line used in two of the figures, this manuscript is certainly publication worthy in its present form.

However, I do have to express disappointment with this study because almost all of the analysis of this wonderful MLS dataset and the sophisticated modeling study that accompanies it is reduced to plots of linear trends. Beyond some lines drawn through two figures (which are dominated by annual cycles) the reader is left with no sense of how well a linear trend actually fits this data. The authors use a quite complicated regression fit that includes some quite long-period terms such as ENSO and 11-year solar cycle terms. Before comparing the trend fits between model and measurement, it would be good to know how well these agree or whether they are significant, since differences in these terms could influence the calculated trends. If they don't make a difference please say so.

A few simple measurement (and possibly model) timeseries plots of annual ClO and HOCl anomalies (without ENSO, F10.7, or QBO fits) or something similar from 50S to 50N at a few of the altitudes shown in Figure 10 would be of great interest. It would visually help the reader to understand how easy it is to identify a linear trend in this data, would help to answer the question of the importance of the multi-year terms in the fit, and would provide some indication of the importance of endpoints. At present, only Figures 2 and 8 provide any timeseries information, and these are very cluttered and difficult to read, dominated by the annual cycle, and given only for very specific (for some reason different for the 2 species) locations.

*Answer: This is a reasonable discussion point (or disappointment), and we agree that showing some simpler time series of deseasonalized anomalies as a top-level view would indeed be a good addition to the Figures already shown (with associated residuals to be shown in the supplementary material, not to add too much in the main portion of the paper). An example of deseasonalized time series is shown below for near-global ClO anomalies at 2 upper stratospheric pressure levels; the QBO signal dominates, after the removal of the main (annual and semi-annual) cycles.*

*In terms of the regression fits, we have used this functional form for other studies where the ENSO term, for example, is more important (for ozone or carbon monoxide in the upper troposphere). However, the ENSO and solar terms do not have a large impact in this upper stratospheric study of ClO and HOCl. Thus, we added a comment to this effect (last 13 lines before Sect. 4), namely:*

In terms of the time series variability and the regression fits, the largest components are, by far, the annual and semi-annual terms (with their relative impacts somewhat dependent on latitude and pressure). For both the observed and modeled near-global cases shown in Fig. 10, about 70–80% of the explained variance arises from these two terms. The ENSO and solar terms typically account for less than a few percent of the explained variance, and the same is true for short-period (less than 6-month) terms. The QBO signal is generally the largest component that remains, if one considers near-global deseasonalized percent anomaly time series, as seen in Fig. 11 for ClO and HOCl at two upper stratospheric pressure levels. The data and model fits generally behave in similar ways, although there can be some small differences between the two. The correlation coefficients between observed HOCl and ClO anomaly time series are of order 0.6-0.7 in the upper stratosphere (with values close to 0.8 if one smoothes out some of the short-term variability in the time series first). The model ClO anomalies track the observed anomalies quite well (with correlation coefficients close to 0.8). We provide the percent residuals associated with Fig. 11 in Fig. S4; these tend to be about twice as large for HOCl (of order $\pm 10\%$) as those for ClO (of order $\pm 5\%$).

[Figure]

**Figure 11.** Deseasonalized anomaly time series (percent) of MLS (blue) and WACCM (red) 50°S–50°N averages over the period 2005 through 2020 for (a) ClO at 3.2 hPa, (b) ClO at 6.8 hPa, (c) HOCl at 3.2 hPa, and (d) HOCl at 6.8 hPa. The linear components of the multivariate linear regression fits are given by dark grey and orange lines for MLS and WACCM, respectively. The associated percent residuals are provided in Figure S4.

*Thus, it is unlikely that the near-global linear trend results would be significantly affected by changes to the regression model (e.g., by using slightly different functional forms or terms), although one can always strive to explain variability in better ways, in order to reduce the error bars to some extent; the linear trends should not be affected very much at all. Such additional analyses would be most useful at 32 hPa, where there is evidence for low frequency (multi-year) ClO variations, although this goes beyond the main purpose of our work (linear trend detection with a focus on the upper stratosphere). However, we did take this general comment to heart, and we added the information mentioned here.*

[Figure]

**Figure S4.** Same as Figs. 2 (for ClO) and 8 (for HOCl), but for residuals and differences of the anomaly time series (percent) shown in Fig. 11 for the 50ºS to 50ºN latitude range, with (a) ClO at 3.2 hPa, (b) ClO at 6.8 hPa, (c) HOCl at 3.2 hPa, and (d) HOCl at 6.8 hPa. The curves have the same meaning as in the case of the residuals and differences from Fig. 2 and Fig. 8.

Figure 2 and Figure 8 – I'm afraid that I don't understand the meaning of "the model fit to MLS data". Is this just WACCM minus MLS? The authors seem to be using "fit" to refer to the regression. Line 246 seems to suggests that the pink line is just a difference: "WACCM time series actually fit the MLS data better than the regression fits do". I indeed hope that this line is just a model minus MLS difference with the bias removed, since this would seem to be the most useful and basic thing to plot.

***Answer:*** *Agreed, the "debiased model minus MLS" wording should have been used for more clarity, so we have changed this accordingly in these two Figures (legend and caption).*

**Minor points:**

Line 138 – "only" is an unnecessary word here. Also, somewhere in this paragraph it should be mentioned that additional details about the standard HOCl retrieval are included in the MLS data quality document (i.e., Livesey et al. 2020). I realize that is mentioned in 3.2.

*Answer: Yes, these minor points have been taken care of accordingly, even though we are focusing on the non-standard HOCl retrieval in this work. See lines 138-139.*

Lines 178-193, or thereabouts- I'm pretty sure that "the model" is always referring to WACCM, but it would be nice to spell that out somewhere in this long paragraph.

*Answer: Indeed, the model is always referring to WACCM; we have added a sentence to make this even more clear (lines 175-176).*

; any reference to "model" in this work refers to this WACCM6 scenario (unless otherwise noted, in particular, for a sensitivity study).

Line 211 – Just curious as to why here the fitted component apparently follows a different solar model than that mentioned in 2.2.

*Answer: The solar flux model used in WACCM6 (and mentioned in Section 2.2) is based on a fit to solar flux data sources, so this is WACCM's historical use of daily solar variability at many wavelengths, including those that are needed for photolysis reactions in the photochemical treatment. Moreover, as mentioned by Gettelman et al. (2019), "It should be noted that beginning 1 January 2015, solar forcing data are projections based on historical solar cycles rather than from observations." The fits to the data are based on another approach, which uses the F10.7 monthly average solar flux data, also a "historical" choice for the regression routines used by the first author. Switching solar models (e.g., for the fits) has not been attempted in this work; given the high correlation one expects between these slightly different solar variability treatments, this will have a negligible effect on the trends and their error bars, given also the very small contribution from the solar term in these fits. Also, I would reiterate that both the model and MLS time series are treated the same way, in terms of the regression fits (including the solar term).*

Line 216 – "year-long blocks". I agree that this is probably a reasonable choice, but I'm just curious if the authors have any particular reason for this choice. I certainly don't insist on any change in the manuscript.

*Answer: The reason is, in part, to follow what others have done in the past, but also because it makes sense to preserve some of the interannual variability by doing so.*

**Comment on acp-2021-880**
**Anonymous Referee #2**

*We thank the referee for the thoughtful comments and suggestions; we address these (in italics) in the text below under **"Answer"**, for each point made. Our revised document will clearly include the changes we point to in detail below.*

**Referee comment on "Upper stratospheric ClO and HOCl trends (2005–2020): Aura Microwave Limb Sounder and model results" by Lucien Froidevaux et al., Atmos. Chem. Phys. Discuss., https://doi.org/10.5194/acp-2021-880-RC2, 2021.**

Review of the manuscript ACP-2021-880 by Froidevaux et al. submitted to ACP and entitled "Upper stratospheric ClO and HOCl trends (2005-2020): Aura Microwave Limb Sounder and model results.

This manuscript presents satellite measurements and model simulations over 50S-50N for the time period 2005-2020 for two inorganic chlorine reservoirs, namely chlorine monoxide (ClO) and hypochlorous acid (HOCl). These global measurements characterize the upper stratosphere with a vertical resolution of 3-4 km (ClO) and 5-6 km (HOCl), they have been derived using the Optimal Estimation Method from Aura-MLS radiometric observations that have been consistently gathered over 16 years (i.e., without the hardware failures that have affected other MLS products/channels, e.g., HCl, N2O). Online and offline products are used, and they present a very good sampling, with about 3500 profiles available per product and per day.

This places the authors in a good position for robust trend determinations for difficult targets, especially HOCl, in support of the Montreal Protocol on substances that deplete ozone. To this end, they use a method or approach that accounts for the auto-correlation often present in geophysical data series, and their model further includes proxies for parameters known to affect the abundance of stratospheric tracers (the solar cycle, the QBO, …).

The investigations are supported by model simulations performed by the WACCM model constrained by MERRA-2 meteorological fields. These simulations are available at the observation sampling, and they are analyzed in the same way than the observations to provide ClO and HOCl climatologies and trends. The model is also used to perform a sensitivity study in order to determine the dependence of the simulated HOCl distributions to the assumed kinetics for its formation reaction (ClO + HO2 -> HOCl + O2), these parameters being still insufficiently defined, and possibly responsible for significant satellite-model biases.

All these investigations are conducted with great care and the results are well presented. The manuscript is well organized, and all the figures are clear, self-explanatory and useful. The text is sometimes a bit lengthy, for instance in section 4 (discussion), but this is probably more a matter of taste than a real issue. It is important to note that this study is dealing with inorganic chlorine reservoirs for which only few observations are available (and the measurements are challenging!), its publication would therefore bring original and interesting elements to the community, also in the context of the continuous evaluation of the success of the Montreal Protocol. In my opinion, this manuscript is almost ready for publication, and I only have a few suggestions to bring to the attention of the authors.

**Primary suggestions or questions.**
In the introduction (line 73), the authors remind us that mid-term variability complicates the trend detection in the lower stratosphere. Among others, Strahan et al (2020) have reported about hemispheric asymmetry in stratospheric transport trends and its impact on the distribution of long-lived tracers. This question is absent elsewhere in this manuscript and global trends are consistently reported. Moreover, most of the material/figures presented do not allow the reader to make his/her own opinion about possible upper stratospheric signals that could characterize the observed and modeled ClO and HOCl data sets. The time series are restricted to a Northern/Southern belt for ClO/HOCl (Fig. 2/8). Still, Fig. 3 and 4 exhibit some asymmetric signals, e.g., at 31.6 hPa for the first one, or the "light green bubble" seen for ClO in the Southern hemisphere in the 20-40 S and 2-10 hPa ranges in Figure 4. My bet is that the authors decided to report global trends because it is already challenging enough, especially for HOCl. But still, I would suggest to indicate in the introduction and/or conclusion that there were no signs of asymmetry in these upper stratospheric data sets, or that they were not searched for.

*Answer: We agree that we could add a few more sentences regarding the issues of asymmetry, given what we already can actually see in the trend Figures that are provided (and these trends follow the time series variations). Doing more than this would require further work, and maybe more importantly, such studies are probably better pursued with time series from longer-lived species (and into the lower altitude portion of the stratosphere, which could not be done in this work on MLS ClO and HOCl). Indeed, underlying lower stratospheric variations in these short-lived species will be significantly influenced by variations in longer-lived species such as $CH_4$ and $H_2O$. At present, we cannot say too much more, but in section 3.1, end of the $2^{nd}$ paragraph, we have added the following sentences:*

We note (from Figs. 3 and 4) that there is some asymmetry in the stratospheric ClO trends between the two hemispheres, with stronger decreases at northern than at southern midlatitudes, and with a somewhat more pronounced effect in the lower stratosphere. However, these asymmetries do not carry much statistical significance. These tendencies are opposite to what has been observed in HCl column trends (see Strahan et al., 2020), which show stronger declines in the south than in the north. Lower stratospheric ClO trends are likely to also be related to trends in $CH_4$ and $H_2O$, although we do not pursue this quantitatively here, other than through the WACCM results, which show a similar but slightly stronger interhemispheric asymmetry in lower stratospheric ClO trend than in the MLS data. At 32 hPa, we note that there is evidence for low frequency (multi-year) MLS and model ClO variations with poorer regression fits to both data and model (although not shown here and not the focus of this work); this complexity is a likely reason for the larger trend discrepancies (WACCM versus data) in this region. Further investigations of interhemispheric asymmetries in lower stratospheric trends (and related age of air issues) are probably best pursued through detailed studies of longer-lived species than ClO.

*Also, in the Conclusion section, we have summarized this by adding the following two sentences at the end of the $2^{nd}$ paragraph:*

Between 15 and 32 hPa, there are indications of some interhemispheric asymmetry in the MLS ClO trends, with faster decreases at northern than at southern midlatitudes, although this is not statistically significant; there is also evidence for low frequency (multi-year) variability, especially at 32 hPa. Further investigations of interhemispheric asymmetries in lower stratospheric trends (and related age of air issues) are probably best pursued through detailed studies of longer-lived species than ClO.

*We did not feel that we could just add two sentences in the Conclusion section without having discussed this, even briefly, in the text. We thank the referee for this suggestion.*

In section 2.2 (line 190), it is indicated that the WACCM6 runs have been augmented to cover the more recent years than the initially available simulations. But since the boundary conditions for the halogenated source gases have been provided by a reference published a few years ago (Meinshausen et al., 2017), I wonder which data have been used in order to describe the post-2016 evolutions of the source gases? A more recent reference is probably needed here.

*Answer: For completeness, we have added a reference (Meinshausen et al., 2020) to the list at the place mentioned by the referee, but more importantly, we note that the CMIP6 scenario is used to project GHG and organic halogen inputs for the model beyond 2014, so this added (underlined) wording should clarify the approach, to a large extent regardless of exact references, or reference dates. See lines 194-195.*

In section 4 (starting line 505), the authors indicate that changes in upper stratospheric temperatures should have had little effects on chlorine partitioning, with less than a 1K decrease as observed by Steiner et al. (2020) over the period of interest here. Regarding the partitioning and trends derived from WACCM6, I guess that the (small) temperature change is accounted for thanks to the MERRA-2 meteorological fields? Or in other words, is the temperature trend in MERRA-2 in agreement with the ~1K trend of Steiner et al. (2020)?

*Answer: Yes, the WACCM6 temperatures follow the MERRA-2 inputs, and MERRA-2 assimilates observational inputs discussed by Steiner et al. (2020).*

Figure 3: I would suggest here to swap the axes (latitude for x-axis; trend for y-axis). This way, Figure 3 would report the latitude on the horizontal scale, as is the case for Fig. 1, 4 and 6.

*Answer: Thank you, we have followed this suggestion; we have also changed the similar Figure (S3) for HOCl in the Supplement.*

**Minor point**
In the introduction (line 59), I would also mention the effect of photolysis (in addition to transport and mixing) to explain the conversion from tropospheric chlorine into the reservoirs.

*Answer: Agreed, we have added the word "photolysis" to the explanation.*